# Analysis of Alpha Activity Levels and Dependence on Meteorological Factors over a Complex Terrain in Northern Iberian Peninsula (2014–2018)

**DOI:** 10.3390/ijerph17217967

**Published:** 2020-10-29

**Authors:** Miguel Ángel Hernández-Ceballos, Fernando Legarda, Natalia Alegría

**Affiliations:** 1Department of Physics, University of Cordoba, 14071 Córdoba, Spain; f92hecem@uco.es; 2Department of Energetic Engineering, University of the Basque Country, 48013 Bilbao, Spain; f.legarda@ehu.eus

**Keywords:** gross alpha activity, northern Iberian Peninsula, radon, airflow patterns, surface winds

## Abstract

Alpha ambient concentrations in ground-level air were measured weekly in Bilbao (northern Spain) by collecting aerosols in filters between 2014 and 2018. Over this period, the alpha activity concentrations in the aerosol’s samples range from 13.9 µBq/m^3^ to 246.5 µBq/m^3^, with a mean of 66.49 ± 39.33 µBq/m^3^. The inter-annual and intra-annual (seasonal and monthly) variations are analyzed, with the highest activity in autumn months and the lowest one in winter months. Special attention has been paid to alpha peak concentrations (weekly concentrations above the 90th percentile) and its relationship with regional meteorological scenarios by means of air mass trajectories and local meteorological parameters. The meteorological analysis of these high alpha concentrations has revealed two airflow patterns-one from the south with land origin and one from the north with maritime origin-mainly associated with these alpha peak concentrations. Surface winds during representative periods of both airflow patterns are also analyzed in combination with ^222^Rn concentrations, which demonstrated the different daily evolution associated with each airflow pattern. The present results are relevant in understanding trends and meteorological factors affecting alpha activity concentrations in this area, and hence, to control potential atmospheric environmental releases and ensure the environmental and public health.

## 1. Introduction

Radioactivity is naturally present in the air due to cosmogenic radionuclides (14C, 7Be, and 3H), cosmic radiation, and natural radionuclides present in all rocks and soils (^238^U and progenies, ^232^Th and progenies, and ^40^K), and its natural background is increased by anthropogenic sources, such as nuclear/radiological accident, nuclear weapon tests, or the production and application of phosphorus fertilizers [1].

The impact of radioactivity on the environment and population health makes the control of radioactivity levels necessary. In this sense, alpha activity concentration is usually used as a control indicator of the radioactivity levels in air, and it is a common measurement of detecting activity peaks associated with radioactive released to the atmosphere [2]. This activity concentration can be measured counting the number of alpha emissions produced per unit of time and air volume (gross alpha), which are determined from the sampling of atmospheric aerosols. The determination and formation of radioactive aerosols in the atmosphere air are analyzed in detail in Papastefanou [3]. In case of alpha activity concentration in aerosols, their origin can be natural due to radon daughters [3].

The long-term monitoring of gross alpha activity concentrations enables us to study trends and periodic variations and, in combination with meteorological parameters, to identify the main meteorological causes influencing the spatial and temporal variability of the environmental radioactivity levels [4]. The seasonal variation and its link with meteorological parameters such as rainfall, temperature, relative humidity, pressure, visibility, wind speed, and wind direction, which influence the concentration of aerosols radiotracers in the atmosphere, has been analyzed in several sites in Spain. Dueñas [5,6] and Piñero-García [7] in the south, García-Talavera [8] in the west, and Saéz-Muñoz [9] in the east provide information about the spatial distribution and the meteorological conditions driven gross alpha concentrations in surface air in Spain. However, in the northern part of Spain, there are completely different meteorological and climatic conditions, and to the authors’ knowledge, there are no studies analyzing alpha concentrations.

The direct relevance of alpha activity on the effective dose causes the need of analyzing and characterizing these concentrations. Depending on the aerosol’s size, in the respiration process, some aerosol settling occurs in the alveolar spaces [10], and if there are some radionuclides on them, internal irradiation is produced [11]. Due to the inhalation of the short-lived radon decay products, this radionuclide delivers the largest proportion of the natural radiation dose to humans [3]. The alpha energy released by Rn progeny is close to 35 GeV/Bq.

In this paper, we analyze the temporal variation of the radioactive aerosol concentrations by measuring weekly the gross alpha activity over five years (2014–2018) at Bilbao (northern Spain), and due to the potential health impact of alpha peak concentrations, we investigated the meteorological factors controlling them. To this end, the link between local/regional meteorological condition and alpha peak concentrations are examined by means of the calculation of backward trajectories (regional scale) and the temporal variability of surface winds and ^222^Rn measured (local scale). This last analysis is based on the fact that, in normal situations, gross alpha origin in air is mainly explained due to the presence of long-lived daughters of gaseous ^222^Rn that are attached to aerosols after cluster formation [3].

The following research issues are addressed in the present study:Analysis of the temporal variation of alpha activity concentrations in surface air;Dependence of alpha peak activity concentrations on airflow patterns;Analysis of the daily temporal evolution of alpha activity concentrations by means the link between surface wind patterns and ^222^Rn concentrations.

The structure of this paper is the following: Section 2 describes the measurement place, the radioactivity measurements, and the meteorological parameters and data used for the study. Section 3 presents, in a first step, the statistical analysis of the alpha activity concentration, while in a second step, the correlation of high alpha concentrations with regional and local meteorological parameters is investigated. To finalize this results section, the dependence of alpha activity concentrations, by means of ^222^Rn values, on the surface wind conditions is analyzed. In Section 4, the conclusions are shown.

## 2. Materials and Methods

### 2.1. Study Area

Bilbao City (43.26 N, −2.94 W) is placed in the Basque Country region (northern Spain) (Figure 1). Bilbao is about 16 km away from the sea, in a mountainous coastal area at the Gulf of Biscay, and in the narrow valley of the Nervion river. The mountains surrounding Bilbao have low altitude (300–800 m), although 40 km away there are mountains with altitudes of 1500 m.

The climate in Bilbao is warm and temperate, with a significant amount of rainfall during the year. Air masses mainly arrive from the west and north to this area. In the period covered by the present study (2014–2018), the temperature averaged 16.0 °C, while the precipitation was about 980 mm per year.

### 2.2. Alpha Activity Concentration and Radon Measurements.

The concentration of radionuclides in soil in this area is very low based on the European Digital Atlas of Natural Radiation [12] (for ^40^K, uranium, thorium, etc.).

The alpha activity concentrations were measured in a monitoring station sited in Bilbao city (43.26 N, −2.94 W) at 34 m above sea level, on the roof of the Faculty of Engineering (Figure 2).

Atmospheric airborne aerosols were collected using a low volume sampler whose nominal flow rate is 30 L/min. It is important to remark that the aerosol sampling takes seven days. In a week the total volume circulated is about 300 m^3^. This sampler uses a cellulose nitrate membrane filter (diameter 47 mm and 0.8 um pore size) which was replaced weekly. The filter is taken from the sampler and placed in a dryer for six days to allow thoron and radon short-lived alpha progeny to decay, and, then, it is measured in a gas flow proportional counter. The gas flow proportional counter used is LB 770 10-Channel α-β Low-Level Counter by Berthold Technologies GmbH & Co. KG trademark. This device is equipped with 10 detectors. The efficiency of the proportional counter calibrated with ^241^Am is close to 20%. A calibration source was prepared ad hoc with the same type of filter as used. The counting time was 6 × 10^4^ seconds, and the detection limits obtained were between 9.71 µBq/m^3^ and 50.68 µBq/m^3^ and uncertainties, with a coverage factor k = 2, were between 14 ± 6.7 µBq/m^3^.

A high flow sampler and a radiological station are located close to this low flow sampler (Figure 2). The radiological station provides hourly radon activity concentrations in the atmosphere, which have also been used in this study.

### 2.3. Meteorology and Backward Trajectories

Ten-minute intervals of the following surface meteorological parameters, which were measured in the weather station placed close to the sampling station (Figure 2), were also used in the present analysis to characterize the impact of local meteorology on alpha activity concentrations: air temperature (°C), relative humidity (%), pressure (mbar), wind direction (°), wind speed (m/s), and precipitation (mm).

Backward trajectories were calculated and used in the present study. Four kinematic three-dimensional backward trajectories per day during specific sampling periods between 2014 and 2018, computed at 00:00, 06:00, 12:00, and 18:00 UTC, with a run time of 96 hours and with a final height of 100 m above ground level, were calculated by using the hybrid single particle Lagrangian integrated trajectory (HYSPLIT) model [13]. The Global Data Assimilation System National Centers for Environmental Prediction (GDAS-NCEP) model, maintained by the National Oceanic and Atmospheric Administration Air Resources Laboratory (NOAA/ARL) were used as meteorological input.

Working with a large number of trajectories requires the application of cluster analysis [14] to classify them according to their pathways, and hence, to identify the corresponding airflow patterns. The present study uses the cluster methodology implemented in the HYSPLIT model and described in Stunder [15]. This multivariate statistical technique groups trajectories with similar direction and lengths by using the Ward’s minimum variance clustering method to minimize the dissimilarity in the trajectories within each cluster while maximizing the dissimilarity of different clusters.

## 3. Results

### 3.1. Alpha Activity Concentration Characterization: Time Series

The alpha activity concentration values, measured weekly from 2014 to 2018, are plotted in chronological order (Figure 3a). In total, there were 258 data, and they were available in each year, thus guaranteeing the largest statistical sample. The activity concentrations in the aerosol’s samples range from 246.5 µBq/m^3^ to 13.97 µBq/m^3^, with a mean of 66.49 ± 39.33 µBq/m^3^, where the uncertainty is given as the sample standard deviation. This Figure 3a suggests a cyclical variability every year, with low values in the first half of the year followed by an increase and low values at the end of the year. This behavior is only broken in 2015. These alpha activity concentrations are also assessed in a box-and-whisker plot (Figure 3b), which shows that there is a symmetrical distribution of the values, i.e., differences between P75 and P25 with the median (P50) are similar. The average is greater than the P50, which denotes the prevalence of low alpha values, although with large impact of occasional high ones. Using the Grubbs test, the existence of outliers among the registered values has been assessed, concluding that there are no outliers in the present set of data.

Alpha measurements in Figure 3a constitute a time series due to the existence of correlation delays of up to 12 weeks between data [16,17]. The periodogram calculated for this period is shown in Figure 4. A periodogram is a helpful tool used to identify the dominant cyclical behavior of a time series. In the Figure 4, the *X*-axis represents frequency and the Y-axis represents the spectrum. Four main peaks, i.e., periodicities of alpha concentration records, are identified in Figure 4. The first one reveals a one-year period variation (frequency 1.93 × 10^−2^ week^−1^ i.e., 52 weeks), while the second, third and fourth peaks indicate intra-annual variability (frequencies 3.9 × 10^−2^, 6.9 × 10^−2^ and 1.6 × 10^−1^ week^−1^ i.e., 25, 12, and six weeks). These results highly agree with the components of the ^7^Be time series identified and analyzed in previous studies [4,18]

Figure 5 displays the box-and-whisker plots of alpha activity concentrations over the years. This Figure 5 shows that the average (square) and the variability of alpha values (P75–P25) are similar over the years, which can be associated with the minimal change in the source term of alpha activity concentrations during this five-year period. The averages are always higher than P50. The average ranges from 60.2 µBq/m^3^ in 2018 to 72.9 µBq/m^3^ in 2015, and the 90th percentile (P90), which is usually taken as reference of maximum values [19], to avoid the impact of possible outliers, ranges from 108.1 µBq/m^3^ in 2014 to 117.2 µBq/m^3^ in 2015.

The analysis of the intra-annual variability, considering winter (December, January, and February), spring (March, April, and May), summer (June, July, and August), and Autumn (September, October, and November), reports the maximum average value in autumn (Figure 6a), concretely in October (110 µBq/m^3^) (Figure 6b), while the minimum average value is in winter (Figure 6a), with the monthly minimum in February (40 µBq/m^3^) (Figure 6b). These seasonal and monthly variability do not agree with previous studies carried out in Spain [5,8,9], in which the activity maxima are registered in summer. This difference would point out differences in source terms and in the meteorological conditions driven alpha activity concentrations during the years among these sites, and hence, the need to analyze the link between alpha and meteorological parameters.

### 3.2. Meteorological Characterization of the Alpha Peak Activity Concentrations

This section presents the results of studying the impact of meteorological parameters on the highest alpha activity concentrations registered in Bilbao for the five years analyzed (2014–2018). The reason for taking the highest alpha activity concentrations as reference is due to health hazards associated with the presence of radionuclides in air [20].

High alpha surface concentrations are identified as values exceeding the P90 calculated over 2014–2018 (117.1 µBq/m3 in Figure 3b). P90 has been previously used in other research as the extreme criterion [18,19,20,21]. The average of the 26 high alpha values identified during the period 2014–2018 above P90 is 152.0 ± 6.6 µBq/m^3^, while the highest alpha value is 246.5 µBq/m^3^. Seasonal distribution of these alpha peak concentrations is shown in Figure 7, as well as the corresponding seasonal mean. This Figure 7 reports the largest occurrence of these high alpha concentrations in autumn (14 out of 26 periods, 53%), and the minimum occurrence in summer (three out of 26 periods, 11.5%). However, the maximum seasonal average of these peak concentrations is registered in spring (187.9 µBq/m^3^), followed by summer (166.7 µBq/m^3^), and lower and similar values in winter (142.4 µBq/m^3^) and autumn (144.8 µBq/m^3^). This seasonal variability of alpha peak concentrations, mainly explained by differences in atmospheric factors, is different to Figure 6a but similar to the seasonal variations shown in previous articles carried out in Spain, and, at the same time, it provides insight into the large impact of occasional high alpha values during the warm (spring/summer) period of the year.

The evaluation of correlation between weekly alpha peak activity concentrations and arithmetic values of local meteorological parameters (temperature, relative humidity, atmospheric pressure, and wind speed and direction) during each sampling period is performed at the significance level of 0.05. In the case of precipitation, the total amount during each sampling period is used. The correlations found are around zero, i.e., there is no relationship between the two variables used. The highest is −0.22 with precipitation, which, being negative, indicates that precipitation tends to reduce alpha activity at the site as a consequence of scavenging of airborne radionuclides by rainfall [22]. These results, with none of them significant at 0.05, indicate the weakness of the relationship between alpha peak concentrations and each individual meteorological variable at this weekly temporal scale level. Therefore, other atmospheric processes should be included in the present analysis aiming at understanding the meteorological reasons of these high alpha surface concentrations in Bilbao.

Due to the impact that airflow patterns have over the surface weather conditions and aerosols temporal distribution in a given area and over a period of time [23], backward trajectories in Bilbao are calculated and are used to determine the influence of regional meteorological patterns on alpha peak concentrations. Figure 8a shows the mean trajectories (cluster) for Bilbao at 100 m calculated considering all the 26 sampling periods above P90. Seven airflow patterns are identified by applying the cluster methodology explained in Stunder [15]. Taking into account the pathways followed by each airflow pattern as well as the frequency (number in brackets) of each one, the arrival of slow continental southern (27%, cluster 4) and maritime northern (28%, cluster 1) air masses dominates periods with high alpha concentrations against the arrival of fast northern (7%, cluster 6) and different Atlantic advections from the northwest (24%, clusters 2 and 3) and from the west (14%, clusters 5 and 7) (Figure 8a).

To establish the link of these airflow patterns with alpha peak concentrations, we have also calculated the airflow patterns associated with the lowest alpha concentrations registered in Bilbao (26 periods in which alpha concentrations were below the 10th percentile) (Figure 8b). The comparison of both (Figure 8) points out similarities between airflow patterns, but with notable differences in frequencies (number in brackets) in those similar. The lowest alpha concentrations highlights the higher occurrence of western airflows, and air masses from the south and north, with slow displacement (Cluster 1 and Cluster 4 in Figure 8a), are not identified. Therefore, both Figures mainly suggest that the arrival of continental southern and maritime northern airflows is mainly associated with measurements of high alpha concentrations in this area.

To study the relationships between airflow patterns (Figure 8a) and alpha peak concentrations, and with the purpose to not consider the same sampling period within two different airflow patterns, which would create misunderstanding in the results, we have identified persistent sampling periods, i.e., periods in which most of the trajectories (above 75% of the calculated trajectories in each period) are grouped into the same cluster. Ten out of 26 sampling periods are identified as persistent ones: six within cluster 1, one within cluster 2, and three within cluster 4. Figure 9 shows the summary of alpha peak concentration measured and the seasonal distribution of these persistent sampling periods. They occurred in autumn/winter seasons in cluster 1 and cluster 4, while the only persistent period corresponding to cluster 2 took place in spring. This result, hence, reports that sampling periods characterized by prevailing southern and northern airflows seems to drive the occurrence of alpha peak activity concentrations in this area. The fact of mainly being located in autumn/winter, which is the wettest period in Bilbao (see Section 2.1), would not be in line with previous studies [6], where alpha activity concentrations decrease with precipitation. However, and due to the geographic location of the region, winds coming from the south descending from the Cantabrian Mountains became warm and dry because the Foehn effect [24], and even within the wettest period, under the arrival of southerly winds, the chances of rain are nearly zero in the region even.

### 3.3. Case Studies

Case studies are needed to evaluate on a real basis the potential influence of each airflow pattern and the associated meteorological conditions on the alpha peak concentrations in Bilbao. So, we have analyzed representative periods of the two main meteorological scenarios driving these concentrations, i.e., the arrival of continental southern and maritime northern air masses. To analyze the influence of each airflow pattern on alpha activity concentrations, Figure 10 displays surface winds (blowing from) during one persistent sampling period (02−08/10/2018) under northerly flows, and from 9 December 2015 to 14 December 2015 under southerly flows. Hourly evolution of ^222^Rn is also shown in both periods to represent the impact of surface winds on activity concentrations.

The alpha activity concentration recorded in the first sampling period is 122.5 µBq/m^3^. According to the wind direction variability, two different periods can be identified in these seven days of sampling (Figure 10a). This fact allows considering this sampling period as suitable to analyze differences in surface winds under the same synoptic meteorological scenario, and its relationship with surface activity concentrations. In addition, at the end of this sampling period was constantly raining.

The first four days, the synoptic conditions favored the progressive development of a daily evolution cycle of surface winds, blowing from the south during the night, followed by the clockwise swing to northwesterly winds during the day. Wind speed was less than 2 m/s. This daily evolution of winds at surface level agrees with the development of sea/land breezes in this area [25], which are more common during spring and summer. Under this surface condition, and considering the marked daily cycle of ^222^Rn [26], it is suggested an accumulation process, which progressively raises the concentration of particles close to the surface in this area, i.e., it traps the particles near the surface and limits a sufficient mixing with the air above [27]. The limitation of the dispersion processes would thus favor the increase in the alpha activity concentration in this area. This daily cycle is broken at the end of the sampling period, with a sudden change in surface winds, which started blowing constantly from the northwest-north range and with wind speeds higher than in the previous days. Under these new conditions, the dispersion processes are favored, and together with the sporadic rainfall during these two days, cause a decrease in activity concentrations in the area, as it well observed in the daily evolution of ^222^Rn concentrations.

The alpha activity concentration recorded in the second sampling period is 149.6 µBq/m3, and there is no rainfall measured in these six days. The synoptic meteorological conditions allowed the constant arrival of surface winds from the south. In this period, changes in wind speed allow identifying two different surface dynamics. In the first five days, the southerly pattern is developed with similar and low wind speeds, which favored similar daily activity concentrations but limited the accumulation processes, not registering high ^222^Rn activity concentrations (comparing with Figure 10a, the radon concentrations differ by about two orders of magnitude). In the last day, the increase in wind speed cause a cleaning effect by enhancing the dispersive effect in the surface atmospheric layers, as can be seen in Figure 10b following the evolution of ^222^Rn activity concentrations.

## 4. Conclusions

Alpha activity concentration can be usually used as a control indicator of the radioactivity levels in air, and so a low-flow station is located in Bilbao, northern Spain. Here, we have analyzed 258 data registered from 2014 to 2018. These alpha activity concentration data can be considered as a time series after autocorrelation function confirms its behavior and periodogram shows an annual periodicity and a significant seasonal and monthly variation of alpha activity concentrations.

In order to explain the alpha peak concentrations, the influence of the airflow patterns and local meteorological parameters has been studied. Precipitation presented a high negative correlation with alpha values, indicating removal process of airborne and the decrease in concentrations. Slow air masses from the south (continental displacement) and from the north (Atlantic displacement) have been found as mainly responsible for the alpha peak concentrations in this area. Under the arrival of northern airflows, the wind directions at the surface levels displayed the development of sea-land breezes, with northern and southern winds, while under the arrival of continental airflows, the arrival of southern winds at surface levels dominates. Analyzing the hourly evolution of ^222^Rn together with surface winds, we have identified how the first meteorological scenario favors the accumulation processes and hence drastically increases activities, while the second scenario keeps activity concentrations similar but with lower values than in the previous scenario during the sampling period.

## Figures and Tables

**Figure 1 ijerph-17-07967-f001:**
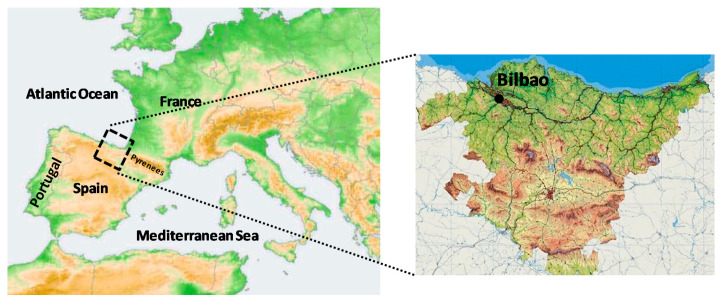
Location of the sampling site in the northern area of the Iberian Peninsula.

**Figure 2 ijerph-17-07967-f002:**
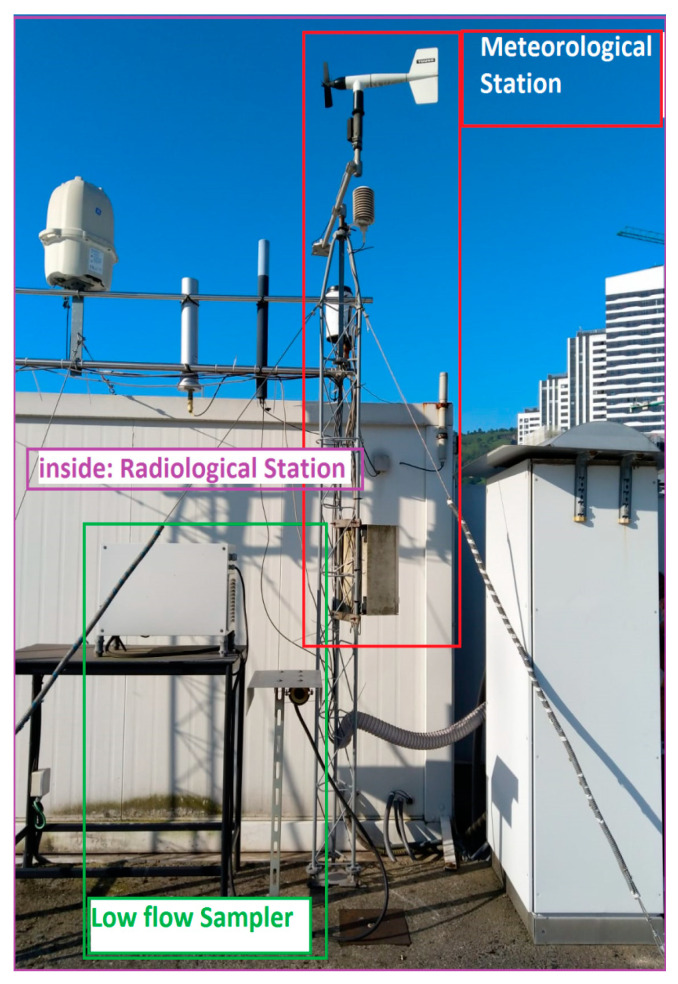
The aerosol sampler, the meteorological station and the radon station used in the present analysis.

**Figure 3 ijerph-17-07967-f003:**
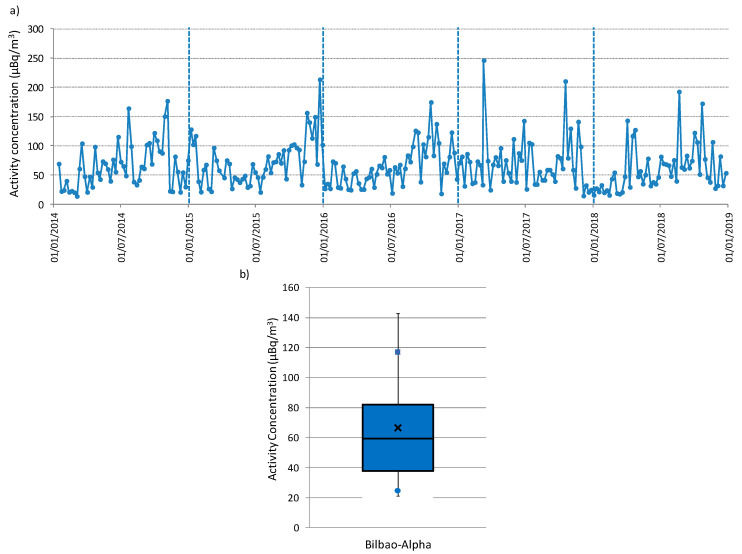
(**a**) Temporal variation and (**b**) box-and-whisker plot of the alpha activity concentrations from January 2014 to December 2018 at Bilbao sampling station. In **b**, the center of the box denotes the 50% (P50), and the bottom and top of the box correspond to the 25% (P25) and 75% (P75) values, respectively. The square indicates the P90 and the circle the P10 values, while the extremes of the box represent the P95 and P5 values. The cross in the box is the average value.

**Figure 4 ijerph-17-07967-f004:**
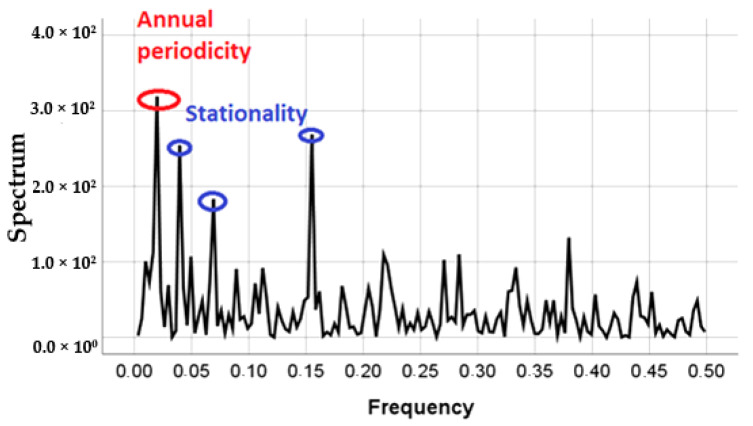
Periodogram for Bilbao time series of alpha activity concentrations from 2014 to 2018.

**Figure 5 ijerph-17-07967-f005:**
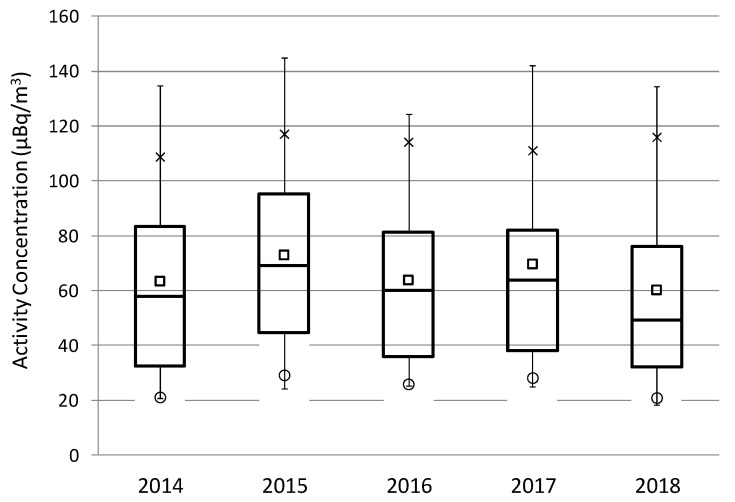
Box-and-whisker plot of alpha activity concentrations over the years (2014–2018) at Bilbao. The meaning of the box plot is the same of Figure 3b.

**Figure 6 ijerph-17-07967-f006:**
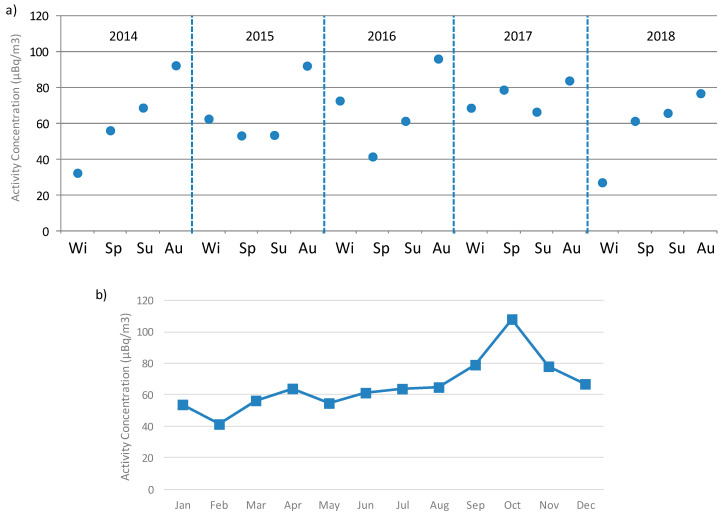
(**a**) Stational variability and (**b**) monthly average of alpha activity concentrations at Bilbao from 2014 to 2018.

**Figure 7 ijerph-17-07967-f007:**
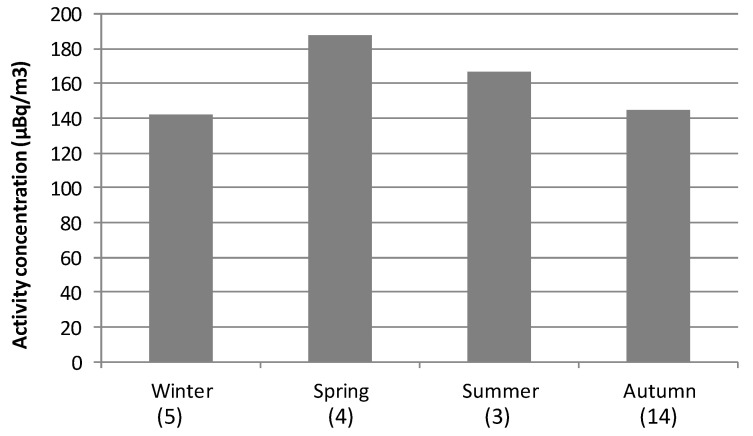
Averaged concentration in each season considering only those sampling periods registering alpha concentrations above P90 during 2014–2018 (in brackets below the season).

**Figure 8 ijerph-17-07967-f008:**
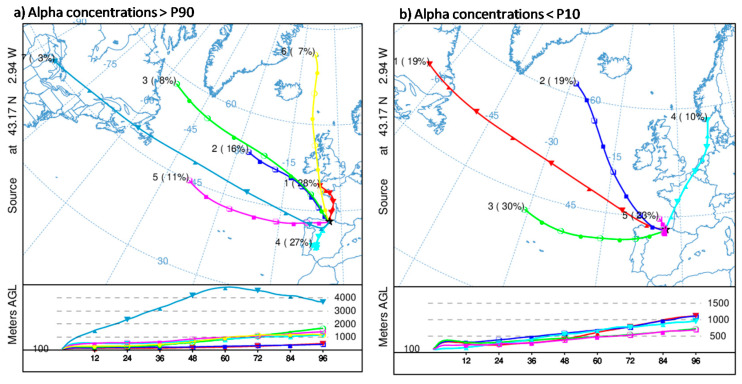
Back-trajectory cluster centers (centroids) obtained at 100 m agl for periods with alpha activity concentrations (**a**) above the 90th percentile and (**b**) below the 10th percentile during the 2014–2018 period at Bilbao. The left numbers in the centroids are an identification number of the centroid and the right numbers (in brackets) are the percentage of complete trajectories occurring in that cluster.

**Figure 9 ijerph-17-07967-f009:**
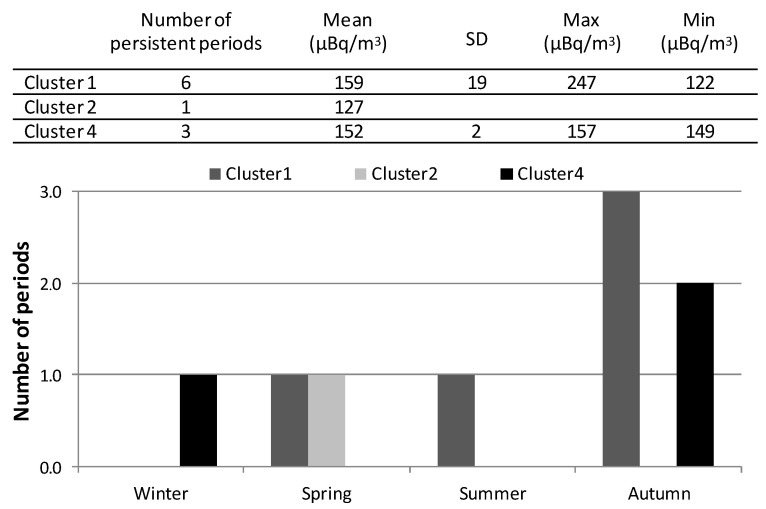
Statistical values in air for alpha peak concentrations during the persistent sampling periods at Bilbao and the corresponding seasonal distribution of the periods. The uncertainties of the means are indicated as the standard deviation of the average Sx/N1/2, where N is the number of persistent periods and the standard deviation. Clusters refer to Figure 8a.

**Figure 10 ijerph-17-07967-f010:**
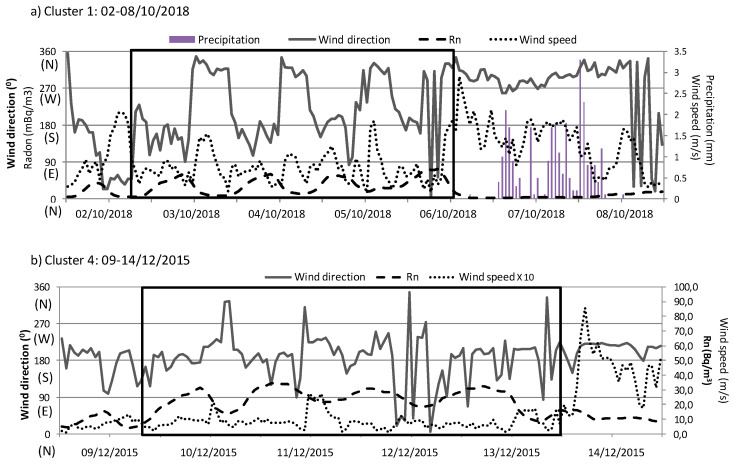
Daily variation of 222Rn activity concentration, wind speed and direction, and total amount of precipitation at Bilbao between (**a**) 2 October 2018 and 8 October 2018 (northerly flows-cluster 1) and (**b**) 9 December 2015 and 14 December 2015 (southerly flows-cluster 4). Note the differences in scales.

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
