# Peer review of "Analysis of Alpha Activity Levels and Dependence on Meteorological Factors over a Complex Terrain in Northern Iberian Peninsula (2014–2018)"

_ijerph, 2020, doi:10.3390/ijerph17217967_

Round 1

Reviewer 1 Report

The analysis of alpha activity is important for the public health. However, there is insufficient information about the measurement system and consideration of the measurement results. Furthermore, there are many minor description errors and the measurement data are unreliable.

Reference No.:

The reference numbers are not in the order in which they appear. Therefore, the consistency between reference number and reference content is questionable.

2.2. Alpha activity concentration and radon measurements: L.98

Is the total sampling volume correct? If the sampling flow rate is 30 L/h in a week, the total sampling volume is calculated to be about 5 m3.

2.2. Alpha activity concentration and radon measurements:

There is no information about the measurement equipment, e.g., company name, the specification and model number or name of device. These information should be described.

2.2. Alpha activity concentration and radon measurements: L 100-103

The authors measured the alpha activity on the filter by a gas flow proportional counter. However, even though the gas flow proportional counter is calibrated by 241-Am, can the gas flow proportional counter measure alpha activity concentration accurately? It must be shown that the results measured by the gas flow proportional counters are only alpha rays caused by 222-Rn and radon daughters on the aerosol.

3.1. Alpha activity concentration characterization: L153-157

The data in figure 4 are not discussed in this article. Does the data have any physical meaning? What did the data reveal? If there is no meaning, the data and text should be omitted.

3.2 Meteorological characterization of the alpha peak activity concentrations: L. 206-207

It is difficult to understand the behavior of alpha activity concentration. What do they mean by "radioactive particles"? Radon daughters are adsorbed and airborne in aerosols, and the aerosols themselves are not radioactive particles. It is necessary to clarify what the alpha activity concentration in the article indicated or contained.

3.3 Case studies:

Define the direction of the wind. Otherwise, the directional descriptions in the text cannot correspond to the graph.

3.3 Case studies:

In the text, the wind speed is stated to be less than 2 m/s. This corresponds to 7.2 km/h, but it is not readily apparent from the graph.

3.3 Case studies:

In the two graphs in Figure 10, the radon concentration differs by about two orders of magnitude. Are the radon concentration data correct?

Author Response

ANSWER TO THE REVIEWERS

Dear Mr Chang:

First of all, we are very grateful to you by the opportunity of publish this paper in International Journal of Environmental Research and Public Health. Thus, we would like deeply acknowledge the work done by the referees in the revision of this manuscript. We appreciate their efforts and contributions to improve the quality of the paper.

In the following paragraphs we describe the response to the comments of the reviewers. In bold-italic letter we have indicated the comments of reviewers, while our answers are in normal style. In the new version of the manuscript, the modifications have been included in blue.

I hope we have met the requirements of reviewer and looking forward to the acceptance of our manuscript in due course of time.

Yours Sincerely,

Dr. Miguel Angel Hernandez Ceballos on behalf of the other authors

Reviewer 2 Report

The manuscript is well written and the experiments lead to results that are clearly described. Those results support the conclusions by the authors.

My only suggestions would be:

1. Include in the Introduction the data regarding the maximum radiation dose that is tolerated by living organisms, especially humans, without any considerable health impairment. This would increase the relevance of this study in terms of rationale.

2. Minor English review is necessary.

3. About one third of the references are from 2015-2020, so it would be best if the authors put some more up-to-date references.

Author Response

(The authors gave the same response as above.)

Round 2

Reviewer 1 Report

The article is well revised. There are no additional comments.